# A role for phagocytosis in inducing cell death during thymocyte negative selection

**Nadia S Kurd[†], Lydia K Lutes, Jaewon Yoon, Shiao Wei Chan, Ivan L Dzhagalov[‡], Ashley R Hoover[§], Ellen A Robey\***

Division of Immunology and Pathogenesis, Department of Molecular and Cell Biology, University of California, Berkeley, Berkeley, United States

**Abstract** Autoreactive thymocytes are eliminated during negative selection in the thymus, a process important for establishing self-tolerance. Thymic phagocytes serve to remove dead thymocytes, but whether they play additional roles during negative selection remains unclear. Here, using a murine thymic slice model in which thymocytes undergo negative selection in situ, we demonstrate that phagocytosis promotes negative selection, and provide evidence for the escape of autoreactive CD8 T cells to the periphery when phagocytosis in the thymus is impaired. We also show that negative selection is more efficient when the phagocyte also presents the negative selecting peptide. Our findings support a model for negative selection in which the death process initiated following strong TCR signaling is facilitated by phagocytosis. Thus, the phagocytic capability of cells that present self-peptides is a key determinant of thymocyte fate.

**\*For correspondence:**
erobey@berkeley.edu

**Present address:** [†]Department of Medicine, University of California San Diego, San Diego, United States; [‡]Institute of Microbiology and Immunology, National Yang-Ming University, Taipei, Taiwan; [§]Oklahoma Medical Research Foundation, Oklahoma, United States

**Competing interests:** The authors declare that no competing interests exist.

## Introduction

During negative selection, thymocytes bearing self-reactive T cell receptors (TCR) are eliminated from the T cell repertoire, an important process for the establishment of self-tolerance. Thymocytes interact with a variety of thymic-resident cells that present self-peptide:MHC complexes, and thymocytes bearing TCRs with high affinity for self-ligands can receive apoptotic death signals (*Starr et al., 2003*). Apoptosis is an immunologically silent form of cell death that is generally thought to be cell-autonomous once initiated (*Strasser et al., 2000*). Although peptide-presenting cells provide the initial apoptotic stimulus to autoreactive thymocytes, whether additional cellular interactions are required to mediate thymocyte death remains unknown.

In addition to the affinity of TCR for self-peptide-MHC, the nature of the peptide-presenting cell is also an important determinant of T cell fate. For example, hematopoietic cells, especially dendritic cells (DC), are potent inducers of negative selection, whereas cortical thymic epithelial cells (cTEC) are specialized to mediate positive selection, promoting thymocyte maturation and survival (*Gallegos and Bevan, 2004*; *Klein et al., 2014*; *McCaughtry et al., 2008*; *Ohnmacht et al., 2009*; *Proietto et al., 2008*; *Taniguchi et al., 2012*; *Wirasinha et al., 2019*; *Yap et al., 2018*). Distinctive features of these cell types, including specialized peptide processing machinery in cTECs and high expression of costimulatory ligands in DCs, play an important role in instructing divergent thymocyte fates (*Klein et al., 2014*; *Starr et al., 2003*). Peptide repertoire and costimulation can contribute to the intensity of TCR signal experienced by a thymocyte, but whether the peptide-presenting cells provide additional TCR-independent signals to promote thymocyte death or differentiation is largely unknown.

A related question is whether a strong TCR signal is sufficient to commit thymocytes to die in a cell-autonomous fashion, or whether other cellular interactions are required. Early observations of apoptotic bodies within thymic phagocytes (*Strasser et al., 2000*; *Surh and Sprent, 1994*), together

with time-lapse microscopy of thymocytes undergoing negative selection (*Dzhagalov et al., 2013*), suggest a close coupling between thymocyte death and phagocytosis. In particular, visualization of dying thymocytes and phagocytes within thymic tissue slices showed that the death of autoreactive thymocytes invariably occurred during close contact with phagocytes, and in most cases death appeared to occur after phagocytosis (*Dzhagalov et al., 2013*). However, it remained unclear whether phagocytes actually contributed to the death of thymocytes by serving as peptide-presenting cells and/or by actively inducing cell death.

Dendritic cells are the most potent antigen-presenting cells (APC) for priming naïve T cells and also present self-peptide for negative selection (*Gallegos and Bevan, 2004*; *McCaughtry et al., 2008*; *Norbury et al., 2002*; *Ohnmacht et al., 2009*; *Proietto et al., 2008*; *Taniguchi et al., 2012*), whereas thymic macrophages are known for their role in clearing away apoptotic thymocytes (*Surh and Sprent, 1994*). Nevertheless, there is considerable functional and phenotypic overlap between DC and macrophages. For example, the marker F4/80 is often used to identify macrophages, but also marks a population with substantial antigen presentation function (*Guerri et al., 2013*). Likewise, the marker CD11c is often used to identify DCs, but is co-expressed with F4/80 by a subset of DC-like cells in the thymus (*Ladi et al., 2008*). To what extent the functions of peptide presentation and phagocytic clearance reside within separate or overlapping thymic cell populations remains unclear.

Phagocytes recognize and uptake apoptotic cells via receptors for 'eat-me' signals displayed on the surface of dying cells (*Arandjelovic and Ravichandran, 2015*). For example, apoptosis induces asymmetry in the plasma membrane, leading to the exposure of phosphatidylserine (PS), which is then recognized by PS receptors on phagocytes. A variety of 'eat-me' receptors are expressed and functional in the thymus (*Elliott et al., 2009*; *Miyanishi et al., 2007*; *Tacke et al., 2015*), but the mechanisms that mediate the efficient removal of autoreactive thymocytes during negative selection have not yet been clearly defined.

Here, we used a thymic slice system in which thymocytes undergo negative selection in situ to address these questions. We show that depletion of thymic phagocytes or blocking phagocytosis impaired negative selection, allowing for the increased survival of thymocytes that experience strong TCR signals. We also identify the PS receptor Tim-4 as an important player during negative selection to tissue-restricted antigens (TRA), and provide evidence for reduced quiescence of CD8 T cells that developed in a Tim-4 deficient environment. Finally, we demonstrate that negative selection is most efficient when the same cell both presents the agonist peptide, and phagocytoses the self-reactive thymocyte. Taken together, our data suggest a model for thymocyte negative selection in which the apoptotic program initiated by strong TCR signals is facilitated by phagocytosis. Thus, thymic phagocytes are not merely 'scavengers', but rather play prominent roles in the induction of self-tolerance, both as peptide-presenting cells and as active inducers of self-reactive thymocyte death.

## Results

### Depletion of thymic phagocytes inhibits negative selection

To examine the role of phagocytes during negative selection, we used thymic tissue slices prepared from Macrophage-Associated Fas-Induced Apoptosis (MAFIA) mice. In these mice, an inducible suicide gene under the control of the colony stimulating factor one receptor (Csf1R) promoter is expressed in DC and macrophage subsets, rendering them susceptible to depletion upon exposure to the small molecule inducer AP20187 (*Burnett et al., 2004*). We have previously observed closely coupled thymocyte death and phagocytosis by GFP$^+$ cells in thymic slices from LysMGFP reporter mice (*Dzhagalov et al., 2013*; *Faust et al., 2000*). Flow cytometric analysis of the thymus of LysMGFP mice revealed that GFP-expressing cells include a subset of F4/80$^{hi}$ macrophages that have been previously described as having potent phagocytic abilities, as well as a subset of CD11c$^{hi}$ DCs (*Figure 1—figure supplement 1*) (*Dzhagalov et al., 2013*; *Tacke et al., 2015*). We confirmed expression of the MAFIA transgene in these subsets using flow cytometric analysis to measure expression of a linked GFP gene (*Figure 1—figure supplement 2a*). We observed that treatment of thymic slices for 16 hr of culture with AP20187 led to the depletion of approximately 50% of F4/80$^{hi}$ macrophages and CD11c$^{hi}$ DC in MAFIA, but not WT, thymic slices (*Figure 1—figure supplement 2b*).

To assess the impact of phagocyte depletion on negative selection, we used a previously described peptide-induced model of negative selection (*Au-Yeung et al., 2014*; *Dzhagalov et al., 2013*). Total thymocytes from mice with a defined MHC class I-restricted TCR transgene (OT-I) were overlaid onto thymic slices with or without the cognate antigen (SIINFEKL, OVAp), along with a reference thymocyte population (either F5 TCR transgenic or polyclonal WT thymocytes) (*Figure 1a*). We used flow cytometry to determine the ratio of viable OT-I:reference thymocytes remaining in the slice as a measure of negative selection (*Au-Yeung et al., 2014*; *Dzhagalov et al., 2013*; *Melichar et al., 2015*) (*Figure 1a*). Consistent with our previous study (*Dzhagalov et al., 2013*), we observed a substantial reduction in the number of live OT-I thymocytes relative to reference thymocytes in thymic slices containing OVAp after 16 hr of culture (*Figure 1b*), with no difference between untreated WT and MAFIA slices (data not shown). In contrast, on phagocyte-depleted slices (MAFIA +AP20187) containing OVAp, the ratio of live OT-I:reference thymocytes was similar to that in control slices without antigen, consistent with the idea that phagocytes promote negative selection.

Thymic phagocytes, including DCs, serve as APCs during negative selection (*Guerri et al., 2013*; *Klein et al., 2014*; *Proietto et al., 2008*; *Taniguchi et al., 2012*). This raises the possibility that the observed defect in negative selection on phagocyte-depleted slices could be the result of thymocytes receiving insufficient TCR signals. However, OT-I thymocytes on phagocyte-depleted slices did not show decreased levels of the TCR activation marker CD69, indicating that surviving thymocytes had received strong TCR signals, in spite of phagocyte depletion (*Figure 1c*). These results suggest that thymocytes on phagocyte-depleted thymic slices exhibit enhanced survival despite continuing to receive strong TCR signals.

## Negative selection and phagocytosis in a model of tissue-restricted antigen presentation

Addition of agonist peptide directly to thymic tissue slices serves as a model for negative selection to ubiquitous self-antigen. To further characterize the relationship between thymocyte death and phagocytosis, we also examined a model of negative selection to TRA. In RIPmOVA transgenic mice, the model antigen ovalbumin (OVA), which provides the agonist peptide for OT-I thymocytes, is expressed in a subset of medullary thymic epithelial cells (mTECs), and is presented in the medulla by mTECs and hematopoietic-derived cells (*Gallegos and Bevan, 2004*; *Kurts et al., 1996*). Because approximately 50% of $CD4^+CD8^+$ double positive thymocytes, and all of the more mature $CD8^+$ single positive thymocytes from OT-I mice express a medullary chemokine receptor pattern ($CXCR4^-CCR7^+$) and migrate to the medulla in thymic slices (*Ehrlich et al., 2009*; *Halkias et al., 2013*; *Kurd and Robey, 2016*; *Yin et al., 2007*), we expected that the majority of OT-I thymocytes would encounter OVA when added to RIPmOVA slices. We observed that approximately 50% of OT-I thymocytes were lost by 9 hr, and there was no further reduction through 48 hr of culture (*Figure 2a*). Thus, the timing and extent of negative selection in a TRA model are in line with our previous results using a ubiquitous model of negative selection (*Dzhagalov et al., 2013*).

We used 2-photon time-lapse microscopy to visualize the interactions between OT-I thymocytes and phagocytes on RIPmOVA thymic slices (*Dzhagalov et al., 2013*). To visualize thymocyte death, we used a previously described method in which thymocytes are double-labeled with a cytosolic dye (SNARF, shown in red) that escapes as cells lose membrane integrity, and a nuclear dye (Hoechst, shown in blue) that increases signal during apoptosis-associated chromatin changes (*Dzhagalov et al., 2013*; *Mempel et al., 2006*). OT-I thymocytes were overlaid onto thymic slices from LysMGFP RIPmOVA transgenic mice and imaged after 6–10 hr of culture. Consistent with our earlier study (*Dzhagalov et al., 2013*), cell death occurred while thymocytes were in close contact with, or engulfed by, GFP-expressing phagocytes (*Figure 2b*, *Videos 1–2*). Thus, the close coupling of autoreactive thymocyte death and phagocytosis occurs during negative selection to both tissue-restricted and ubiquitous self-antigen.

## Phagocyte killing of autoreactive thymocytes is mediated by phosphatidylserine receptors

To determine whether phagocytes exert their effect on negative selection specifically through phagocytosis, we evaluated the efficiency of negative selection in an environment in which phagocytes are present, but impaired in their ability to phagocytose. Phosphatidylserine (PS) is exposed at

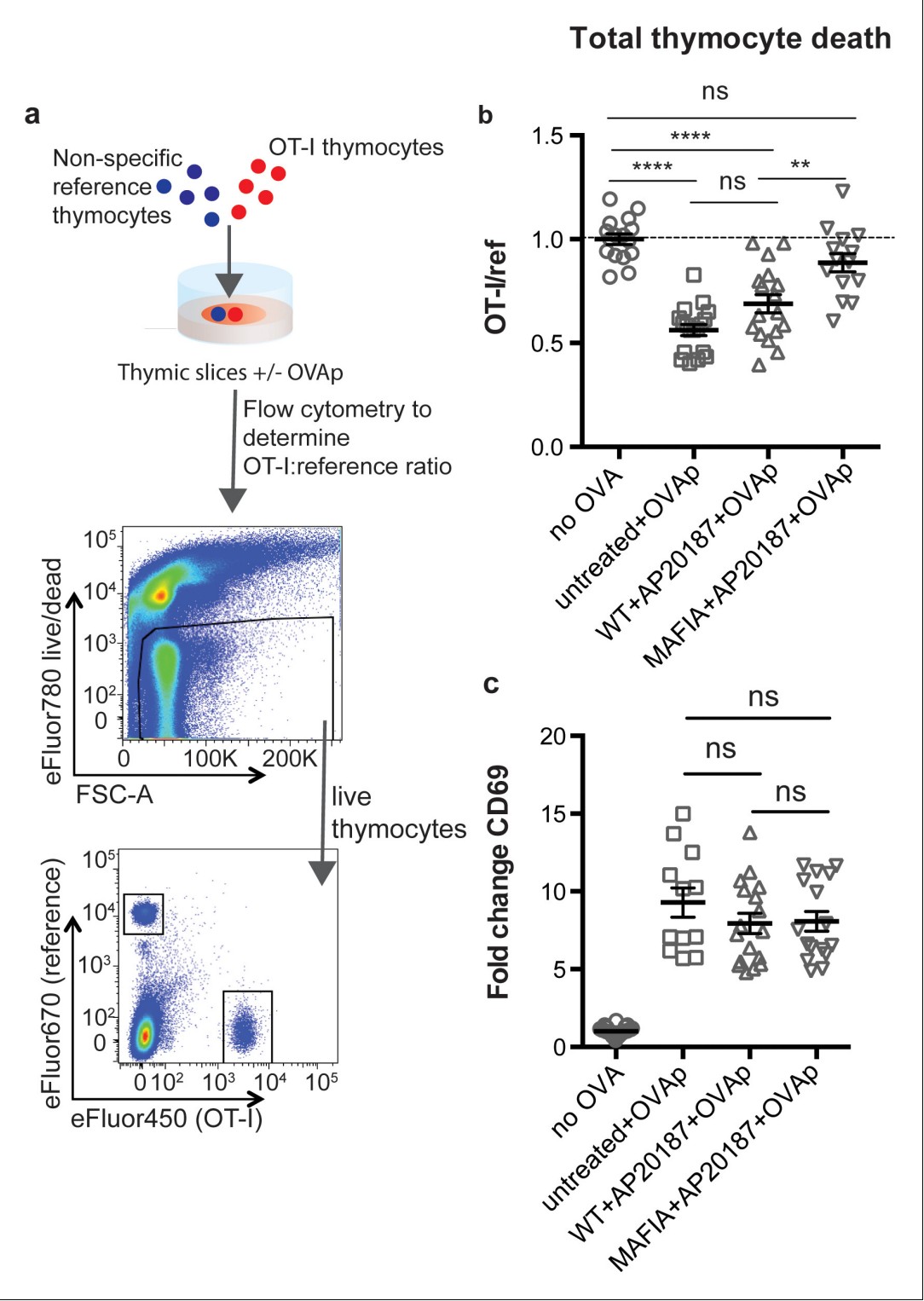

**Figure 1.** Depletion of phagocytes inhibits negative selection without dampening antigen recognition. (**a**) Strategy to quantify negative selection: labeled OT-I and reference thymocytes (either wild type or F5 TCR transgenic) were overlaid onto thymic slices with or without OVA peptide, cultured for 16 hr, and then dissociated for analysis by flow cytometry. Lower panels show the flow cytometry gating strategy used to quantify live OT-I and reference thymocytes. (**b–c**) For depletion of phagocytes, WT or MAFIA thymic slices were treated with AP20187 and cultured for an additional 16 hr prior to the addition of thymocytes and peptide. (**b**) Thymocyte death displayed as the ratio of live OT-I thymocytes relative to live reference thymocytes present within the slice. We further

*Figure 1 continued on next page*

*Figure 1 continued*

normalized the ratios of OT-I:reference thymocytes in each experiment so that the average of the corresponding 'no OVA' samples was 1.0. (c) Fold change in expression of the activation marker CD69 by surviving OT-I thymocytes displayed as Mean Fluorescence Intensity (MFI) normalized so that the average MFI of the corresponding 'no OVA' samples is set to 1.0. ns not significant ($p > 0.05$), **$p < 0.01$, ****$p < 0.0001$ (one-way ANOVA with Bonferroni's correction with a 95% confidence interval) Data are pooled from three independent experiments, with mean and SEM of n = 12–39 total slices per condition, where each dot represents an individual slice.

The online version of this article includes the following figure supplement(s) for figure 1:

**Figure supplement 1.** GFP-expressing cells in the LysMGFP thymus include F4/80$^{hi}$ macrophages as well as a subset of CD11c$^{hi}$ DCs.

**Figure supplement 2.** Depletion of phagocytes in MAFIA thymic slices.

---

the cell surface early in the apoptotic process, and serves as an 'eat-me' signal to phagocytes (*Arandjelovic and Ravichandran, 2015*). To determine whether PS mediates phagocytosis of self-reactive thymocytes, we used Annexin V (AnnV), a small protein that binds to PS, to block the interaction between PS and its receptors in thymic slices (*Krahling et al., 1999*). Negative selection on RIPmOVA thymic slices was completely blocked by treatment of thymic slices with AnnV (*Figure 3a*). This was not due to a failure of thymocytes to receive strong TCR signals, since CD69 upregulation on surviving thymocytes was not impaired (*Figure 3b*). AnnV addition had a similar effect on negative selection in response to OVAp (*Figure 3c,d*). These results confirm that phagocytosis promotes the death of autoreactive thymocytes, and suggest that PS receptors are important for this process.

## The phosphatidylserine receptor Tim-4 promotes negative selection of CD8 T cells in the thymus

Phagocytes express a number of receptors for PS, allowing them to recognize and uptake apoptotic cells (*Arandjelovic and Ravichandran, 2015*). These include Tim-4, a PS receptor previously reported to be expressed and functional in the thymus (*Tacke et al., 2015*). Using flow cytometry, we found that Tim-4 is expressed by almost all F4/80$^{hi}$ macrophages and ~30% of CD11c$^{hi}$ DCs (*Figure 4—figure supplement 1a*). To investigate whether Tim-4 plays a role in negative selection, we prepared thymic tissue slices from *Timd4$^{-/-}$* RIPmOVA transgenic mice, overlaid OT-I and reference thymocytes, and assessed the extent of negative selection after 16 hr of culture. OT-I thymocyte death was not detectable in *Timd4$^{-/-}$* RIPmOVA thymic slices, but was readily detectable in thymic slices from age and sex-matched *Timd4$^{+/+}$* RIPmOVA controls (*Figure 4a*). Negative selection in response to OVAp was also slightly reduced on *Timd4$^{-/-}$* thymic slices, although the difference did not reach statistical significance (*Figure 4c*). Normal CD69 upregulation by OT-I thymocytes suggested that surviving thymocytes received strong TCR signals, even when OVA is presented in the *Timd4$^{-/-}$* thymic environment (*Figure 4b,d*). This is consistent with the normal number and phenotype of thymic phagocytes that we observed in *Timd4$^{-/-}$* mice (*Figure 4—figure supplement 1b,c, d*). Taken together, these data support the idea that phagocytosis promotes effective negative selection, and suggest that Tim-4 is a relevant player in this process.

If Tim-4 mediated phagocytosis plays a role in negative selection, we might expect an increase in the escape of autoreactive T cells in *Timd4$^{-/-}$* mice. Consistent with this, we observed an increase in CD8 T cells with an activated phenotype (CD44$^{hi}$CD62L$^{lo}$) in mature CD8 T cells in the spleens of *Timd4$^{-/-}$* mice (*Figure 5a*). To investigate whether defective phagocytosis in the thymus might contribute to this increase, we isolated thymocytes from *Timd4$^{-/-}$* and wild type mice and transferred them into T cell-deficient (*Tcra$^{-/-}$*) mice. In this setting, mature thymocytes that had developed in an environment where thymic phagocytes lack Tim-4 were allowed to further mature as T cells in a peripheral environment with Tim-4-sufficient phagocytes. While the phenotype of T cells, including the proportion of dividing cells (Ki67$^{hi}$), derived from *Timd4$^{-/-}$* versus wild type mice were similar (*Figure 5b,c* and data not shown), we noted a reproducible increase in the level of Ki67 expression amongst 'Ki67$^{lo}$' CD8 T cells derived from thymocytes of *Timd4$^{-/-}$* mice (*Figure 5b,d*) indicative of more rapid transit through G0 (*Hogan et al., 2013*; *Miller et al., 2018*; *Sobecki et al., 2017*). This difference, which was also apparent in the blood of steady-state *Timd4$^{-/-}$* animals (*Figure 5—figure*

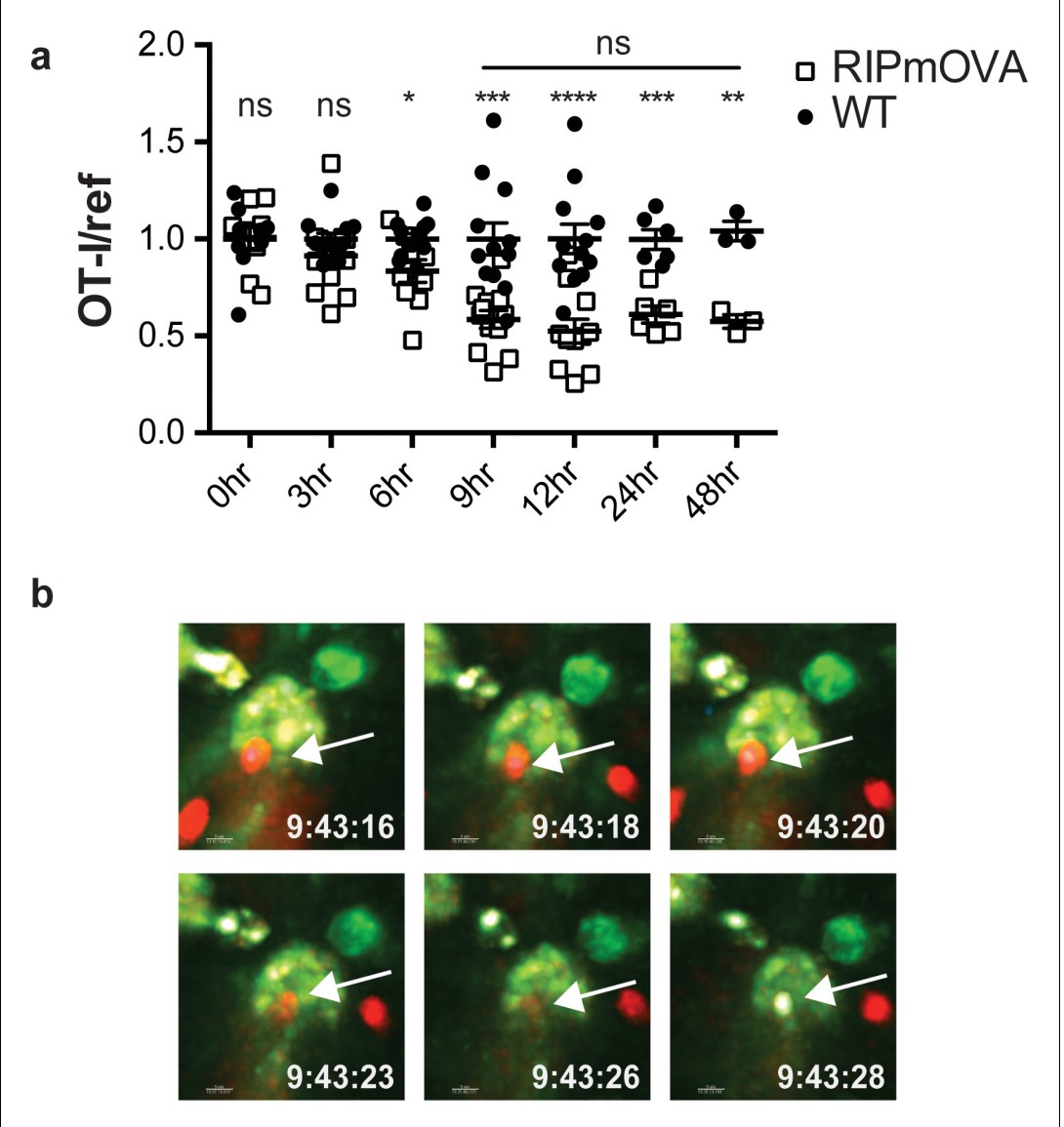

**Figure 2.** Thymocyte death and phagocytosis during negative selection to tissue-restricted antigen. (**a**) Negative selection on RIPmOVA slices (open squares), displayed as the ratio of live OT-I thymocytes relative to live reference thymocytes present within the slice, normalized to no antigen controls (black squares). Data are pooled from 4 (0, 3, 6, 9, and 12 hr timepoints), 2 (24 hour timepoint), or 1 (48 hour timepoint) experiments, with mean and SEM of n = 3 (48hr WT and RIPmOVA), 6 (24hr WT and RIPmOVA), 9 (0hr WT and RIPmOVA), 7 (6hr RIPmOVA), 11 (12hr WT), 12 (3hr and 9 hr WT and RIPmOVA, 6 hr and 12 hr WT) total slices per condition, respectively, where each dot represents an individual slice. ns not significant (p>0.05), *p<0.05, **p<0.01, ***p<0.001, ****p<0.0001. Unpaired two-tailed Student's *t*-test of WT vs RIPmOVA for each timepoint, or two-way ANOVA with 95% confidence interval with Tukey's multiple comparisons test to compare RIPmOVA samples across timepoints (horizontal line). (**b**) Still images from a time-lapse series showing an example of OT-I thymocyte death. OT-I thymocytes were depleted of mature CD8 SP and labeled with Hoechst and SNARF before overlaying on LysMGFP RIPmOVA thymic slices. Slices were imaged by two-photon scanning laser microscopy. A 30 min movie was recorded in the medulla, with the time elapsed since thymocyte entry into the slice shown in white. Arrows indicate the position of the dying thymocyte.

*supplement 1*), together with the increase in activated CD8 T cells in intact *Timd4*[-/-] mice (*Figure 5a*), is consistent with increased self-reactivity of CD8 T cells. Interestingly, these differences were not observed amongst CD4 T cells (*Figure 5a,d*). These data suggest that defective phagocytosis in the thymus of *Timd4*[-/-] mice leads to the escape of CD8 T cells with borderline self-reactivity,

and support the idea that Tim-4-mediated phagocytosis in the thymus facilitates efficient negative selection of CD8 T cells.

## Antigen presentation by phagocytes promotes efficient negative selection

The close association between autoreactive thymocytes and phagocytes just prior to their engulfment and death (*Dzhagalov et al., 2013*) (*Figure 2c*, *Videos 1–2*) suggests that phagocytes may also serve as APCs. Moreover, the ability of a peptide-presenting cell to phagocytose may make it more potent at inducing negative selection. To test this idea, we took advantage of the fact that bone marrow-derived dendritic cells (DCs) have phagocytic activity (*Figure 6—figure supplement 1*) and can serve as exogenous peptide presenting cells when overlaid and allowed to migrate into thymic slices (*Melichar et al., 2013*; *Weist et al., 2015*). We added OVA-loaded *Timd4*-/- or WT DCs onto thymic slices that had been previously overlaid with OT-I and reference thymocytes, and measured the extent of negative selection 16 hr later (*Figure 6a*). Interestingly, while OVA-loaded WT DCs induced a significant level of negative selection, *Timd4*-/- DCs failed to induce detectable negative selection (*Figure 6b*). This defect was not due to defective peptide presentation by *Timd4*-/- DCs, as surviving OT-I thymocytes strongly upregulated CD69 in the presence of OVA-loaded *Timd4*-/- DCs (*Figure 6c*). The fact that peptide-presenting cells defective in phagocytosis were unable to effectively induce negative selection, despite the fact that functional (non-presenting) endogenous phagocytes were present in the thymic slice, indicates that negative selection is more efficient when the same cell serves both as APC and phagocyte.

## Discussion

While the importance of phagocytes in clearing dead thymocytes has long been appreciated, their role during negative selection prior to thymocyte death remained unknown. Here, we provide evidence for key roles for phagocytes as antigen presenting cells and as active inducers of thymocyte death. We demonstrate that phagocytosis promotes autoreactive thymocyte death, and that the recognition of PS is critical for this process. Additionally, we show that negative selection is most efficient when the phagocyte also presents the negative selecting peptide. Taken together, our data support a model for negative selection in which phagocytosis helps to enforce the ultimate death of thymocytes that have initiated the apoptotic process following a strong TCR signal. Moreover, the coupling of these two steps that occurs when phagocytes also serve as APCs leads to more efficient negative selection (*Figure 6—figure supplement 2a*).

In vitro studies of apoptosis support a model in which a death signal initiates a cell-autonomous program of cellular destruction involving protease and nuclease activation, PS exposure on the outer membrane, membrane blebbing, and the formation of apoptotic

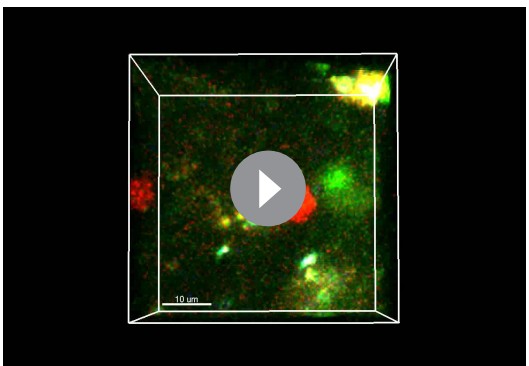

**Video 1.** Example 1 of thymocyte death occurring concurrently with phagocytosis. OT-I thymocytes were depleted of mature CD8 SP and double labeled with Hoechst and SNARF before overlaying on LysMGFP RIPmOVA thymic slices. Slices were imaged by two-photon scanning laser microscopy at 9.5 hr after thymocyte addition to the slice.
https://elifesciences.org/articles/48097#video1

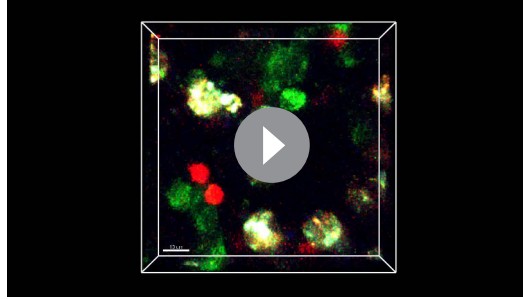

**Video 2.** Example 2 of thymocyte death occuring concurrently with phagocytosis.
https://elifesciences.org/articles/48097#video2

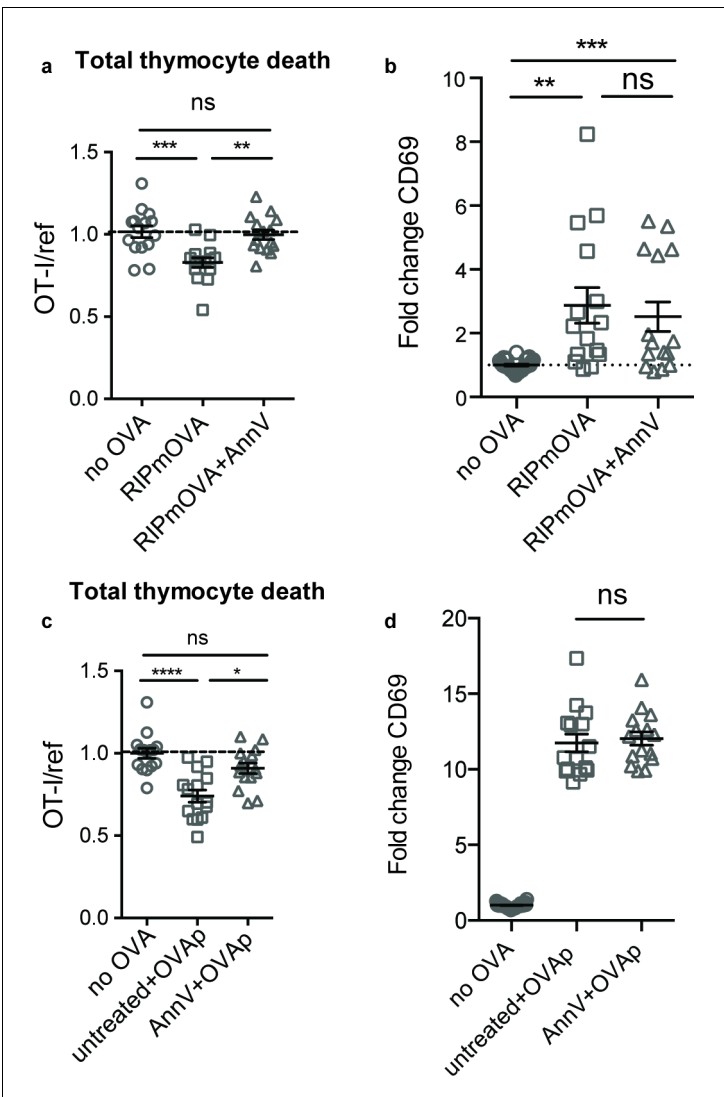

**Figure 3.** Phagocyte killing of autoreactive thymocytes is mediated by phosphatidylserine receptors. (a-d) OT-I and reference thymocytes in AnnV buffer with or without AnnV were overlaid onto RIPmOVA slices (a,b) or WT slices treated with OVAp (c,d). Slices were then treated with AnnV and harvested 16 hr later for flow cytometric analysis. (a,c) Negative selection displayed as the ratio of live OT-I thymocytes relative to live reference thymocytes, normalized to no antigen controls. (b,d) Fold change in expression of the activation marker CD69 by surviving OT-I thymocytes displayed as Mean Fluorescence Intensity (MFI) normalized so that the average MFI of the corresponding 'no OVA' samples is set to 1.0. ns not significant (p>0.05), **p<0.01, ***p<0.001, ****p<0.0001 (one-way ANOVA with Bonferroni's correction with a 95% confidence interval). Data are pooled from three independent experiments, with mean and SEM of n = 15–30 total slices per condition, where each dot represents an individual slice.

bodies (*Strasser et al., 2000*). Our current work adds to a growing body of evidence that the early events associated with apoptosis are not necessarily a death sentence. For example, activated T cells can transiently express active caspase-3 and expose PS at the cell surface, but ultimately avoid cell death (*Alam et al., 1999*; *Elliott et al., 2005*; *Fischer et al., 2006*; *McComb et al., 2010*; *Miossec et al., 1997*). Moreover, phagocyte-dependent killing is critical for regulating the size of T cell, erythrocyte, and neutrophil populations, as well as removal of transient structures during development (*Albacker et al., 2010*; *Dalli et al., 2012*; *Hoeppner et al., 2001*; *Khandelwal et al., 2007*; *Lang and Bishop, 1993*; *Marín-Teva et al., 2004*; *Oldenborg et al., 2000*; *Park et al., 2008*; *Reddien et al., 2001*). Thus, mounting evidence suggests that phagocytosis contributes to the

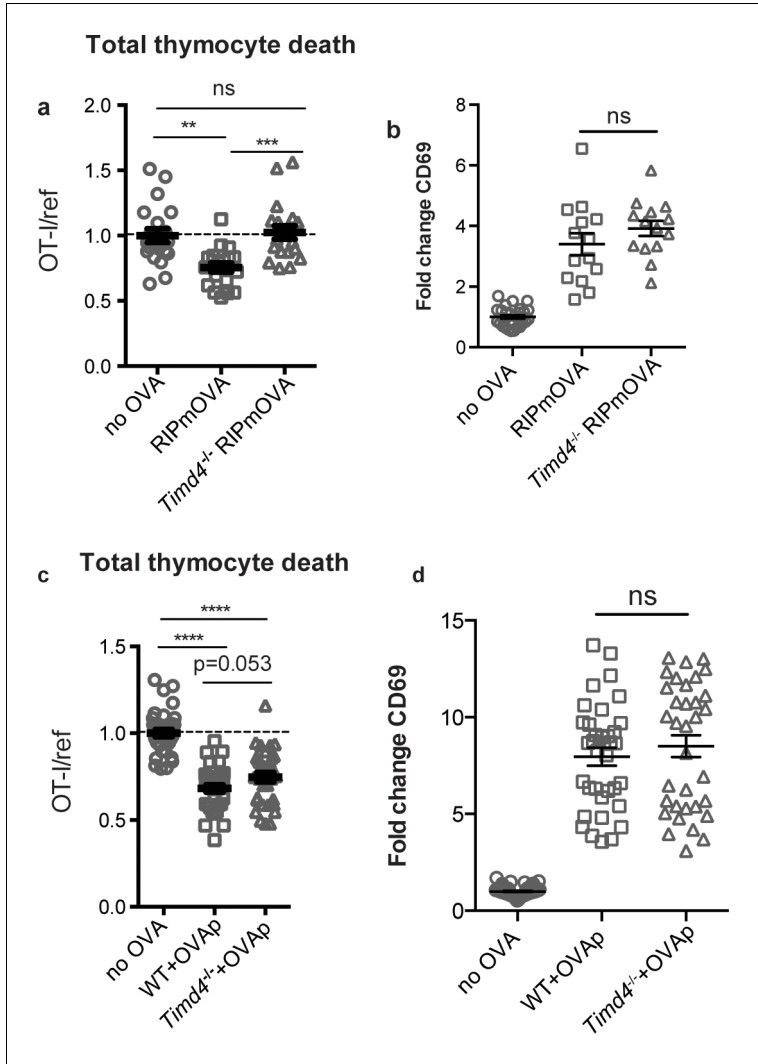

**Figure 4.** The phosphatidylserine receptor Tim-4 promotes negative selection to tissue-restricted antigens. (a–d) OT-I and reference thymocytes were overlaid onto WT or $Timd4^{-/-}$ thymic slices with or without the RIPmOVA transgene. (a,b), or with or without addition of OVAp (c,d), and slices were dissociated and analyzed by flow cytometry 16 hr later. (a,c) Negative selection displayed as the ratio of surviving OT-I thymocytes relative to reference thymocytes, normalized to no antigen controls. (b,d) Fold change in expression of the activation marker CD69 by surviving OT-I thymocytes displayed as Mean Fluorescence Intensity (MFI) normalized so that the average MFI of the corresponding 'no OVA' samples is set to 1.0. ns not significant (p>0.05), **p<0.01, ***p<0.001, ****p<0.0001 (one-way ANOVA with Bonferroni's correction with 95% confidence interval) Data are pooled from 4 (a,b) or 7 (c,d) independent experiments, with mean and SEM of n = 15–20 (a,b) or 35 (c,d) total slices per condition, where each dot represents an individual slice.

The online version of this article includes the following figure supplement(s) for figure 4:

**Figure supplement 1.** Normal number and cell-surface phenotype of phagocytes in the thymus of $Timd4^{-/-}$ mice.

---

extent of cell death in vivo. While it is possible that autoreactive thymocytes that are not phagocytosed would ultimately undergo a delayed death, our data indicating that CD8 T cells that develop in a Tim-4 deficient environment display reduced quiescence support the view that at least some of these cells may escape deletion and persist as mature T cells with increased self-reactivity. It is also possible that some thymocytes that escape phagocytosis and go on to give rise to alternative T cell lineages, such as CD8αα intraepithelial lymphocytes (*Leishman et al., 2002*; *Mayans et al., 2014*; *McDonald et al., 2014*).

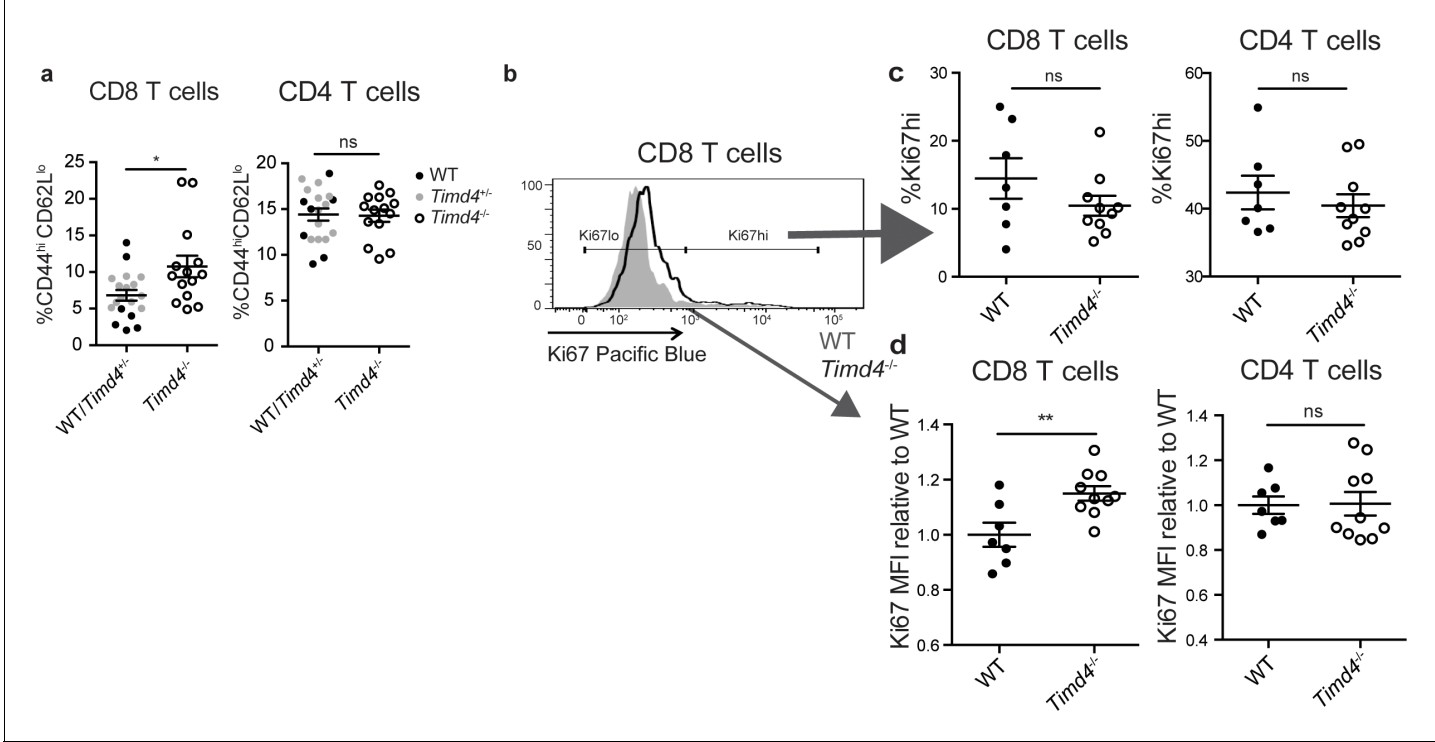

**Figure 5.** CD8 SP thymocytes that develop in a Tim4-deficient environment show signs of increased self-reactivity. (**a**) Percent of CD8 (left) or CD4 (right) T cells with an activated phenotype (CD44$^{hi}$ CD62L$^{lo}$), isolated from the spleens of age and sex-matched WT and *Timd4*$^{-/-}$ mice, or *Timd4*$^{-/-}$ and littermate *Timd4*$^{+/-}$ controls. Data are compiled from n = 14 *Timd4*$^{-/-}$ mice and n = 18 WT or *Timd4*$^{+/-}$ controls, where each dot represents an individual mouse. (**b–d**) Thymocytes from age and sex-matched *Timd4*$^{-/-}$ or WT mice were injected i.v. into *Tcra*$^{-/-}$ hosts. Blood was collected 9–10 weeks post-transfer and analyzed by flow cytometry. (**b**) Ki67 expression of CD8 T cells derived from *Timd4*$^{-/-}$ or WT mice. (**c**) Percent of CD8 or CD4 T cells expressing high levels of Ki67. (**d**) Relative levels of Ki67 within resting (Ki67$^{lo}$) WT or *Timd4*$^{-/-}$ CD8 or CD4 T cells, represented as MFI normalized to WT levels, where each dot represents an individual mouse. Data are representative of (**b**) or pooled from (**c,d**) two independent experiments, with mean and SEM of n = 7 (WT) or 10 (Tim4$^{-/-}$). ns not significant (p>0.05), *p<0.05, **p<0.01 (two-tailed Student's *t*-test with 95% confidence interval).
The online version of this article includes the following figure supplement(s) for figure 5:

**Figure supplement 1.** Elevated levels of Ki67 in *Timd4*$^{-/-}$ CD8 T cells at steady-state.

The mechanism by which phagocytes actually kill their target cells remains unknown. Following phagocytosis, the engulfed cell would be exposed to lysosomal proteases and other degradative enzymes. Furthermore, under certain conditions phagocytes release reactive oxygen species (ROS) into the phagosome, and ROS have been shown to be important for the phagocytosis-induced death of neurons during development in the mouse brain (**Marín-Teva et al., 2004**). It is tempting to speculate that the cytotoxic environment that a cell encounters within the phagosome following engulfment is ultimately responsible for its death. Although indirect mechanisms such as retinoic acid and cytokine production by phagocytes (**Sarang et al., 2013**) and PS signaling (**Elliott et al., 2005**), might also impact thymocyte survival, the fact that we observe similar defects in negative selection upon phagocyte depletion, blocking PS, and mutation of a PS receptor, strongly suggests that phagocytes mediate thymocyte death directly via phagocytosis.

Our observation that negative selection is most efficient when phagocytes also serve as peptide-presenting cells implies that the phagocytic activity of a peptide-presenting cell is a key factor in determining thymocyte fate. Indeed, the inability to phagocytose could help to explain the relative inefficiency of thymic stromal cells to induce negative selection (**McCaughtry et al., 2008**; **Melichar et al., 2013**; **Ohnmacht et al., 2009**; **Proietto et al., 2008**; **Wirasinha et al., 2019**; **Yap et al., 2018**), in contrast to the phagocytic activity of many hematopoietic cells which are potent inducers of negative selection. It is worth noting that many thymic phagocytes, identified in this study by the expression of the LysM-GFP reporter, also express the DC marker CD11c (**Ladi et al., 2008**) (**Figure 1—figure supplement 1**). Thus, a subset of thymic hematopoietic cells with

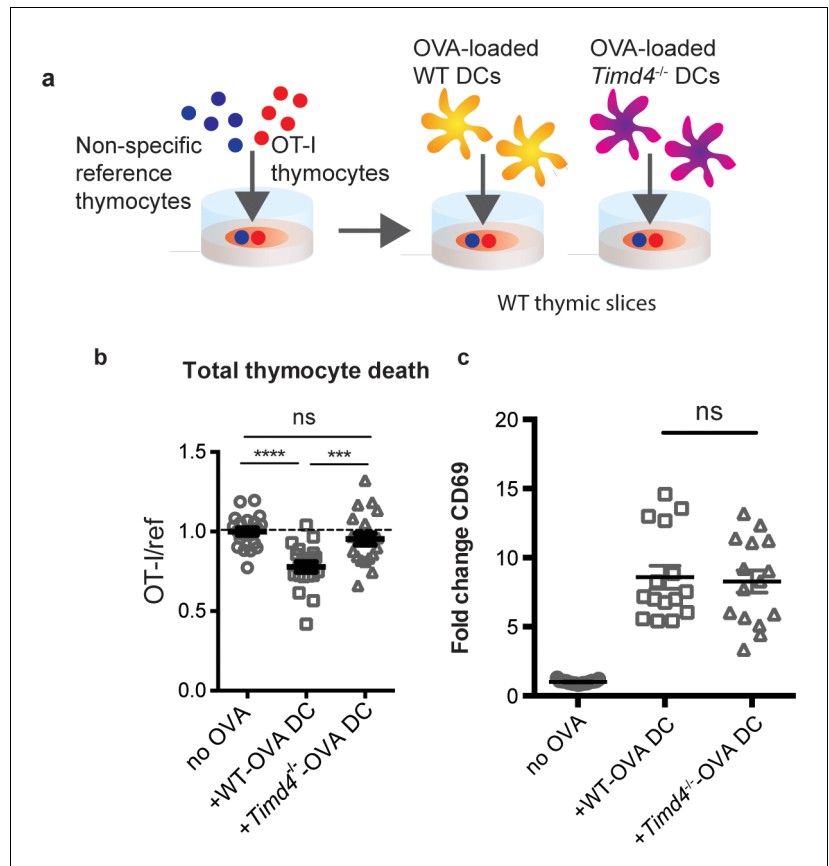

**Figure 6.** Peptide presentation by phagocytes promotes efficient negative selection. (**a**) Schematic of the experimental setup: OT-I and reference thymocytes were overlaid onto WT thymic slices onto which OVA-loaded or unloaded WT or Tim-4[-/-] BMDCs were added. Slices were dissociated and analyzed by flow cytometry 16 hr later. (**b**) Negative selection displayed as the ratio of live OT-I thymocytes relative to live reference thymocytes, normalized to no antigen controls. (**c**) Antigen recognition of surviving OT-I thymocytes displayed as Mean Fluorescence Intensity (MFI) of the activation marker CD69. Data are pooled from 4 (**b**) or 3 (**c**) independent experiments, with mean and SEM of n = 15–20 total slices per condition, where each dot represents an individual slice. ns not significant (p>0.05), ***p<0.001, ****p<0.0001 (one-way ANOVA with Bonferroni's correction with 95% confidence interval).

The online version of this article includes the following figure supplement(s) for figure 6:

**Figure supplement 1.** Bone marrow-derived dendritic cells are phagocytic.
**Figure supplement 2.** Model for the role of phagocytes in inducing cell death during negative selection.

characteristics of both DCs and macrophages may be particularly important mediators of negative selection. Although some non-phagocytic subsets, including mTECs, thymocytes, and B cells, can induce negative selection (*Gallegos and Bevan, 2004*; *Melichar et al., 2015*; *Yamano et al., 2015*), this process may be rendered less efficient by the requirement for a second cellular encounter with a phagocyte in order to complete negative selection (*Figure 6—figure supplement 2b*).

The initiation of phagocytosis is dependent on the recognition of target cells via a variety of receptors for 'eat me' signals, including PS. We found that mutation of a single PS receptor, Tim-4, impaired negative selection to TRA, but had a less obvious impact on negative selection to exoge-nously added antigenic peptide. On the other hand, globally blocking PS significantly impaired both forms of negative selection. These results could indicate that Tim-4 plays a non-redundant role in phagocytosis in the medulla, the site of TRA presentation, whereas other PS receptors may partici-pate in phagocytosis in the thymic cortex, where negative selection to ubiquitous antigen takes place (*Daley et al., 2013*; *Klein et al., 2014*; *McCaughtry et al., 2008*). Alternatively, negative selection to TRA might be more sensitive to perturbations in phagocytosis, given that TRA are

typically expressed at low levels, and may deliver a relatively weak apoptotic signal. In line with this idea, cells receiving a weak apoptotic signal in *C. elegans* embryos are more dependent upon phagocytosis for their death than cells receiving a strong apoptotic signal (*Hoeppner et al., 2001*). This raises the intriguing possibility that phagocytes might be especially critical during negative selection to relatively low-affinity or rare self-antigens, which pose the greatest risk as targets of autoimmunity (*Koehli et al., 2014*).

A requirement for Tim-4 in negative selection involving relatively weak apoptotic stimuli is consistent with the mild autoimmune phenotype reported in *Timd4*$^{-/-}$ mice (*Rodriguez-Manzanet et al., 2010*). Although their hyperimmune phenotype was initially attributed to defective phagocytosis in the periphery (*Albacker et al., 2010*; *Rodriguez-Manzanet et al., 2010*), our data indicate that CD8 T cells that arise in a Tim-4 deficient thymic environment are less quiescent compared to normal CD8 T cells. Specifically, we observed that CD8 T cells that developed in a thymic environment lacking Tim-4, but then further matured in a Tim-4 sufficient peripheral environment, exhibited slightly elevated levels of Ki67. Although Ki67 is widely used as a binary marker of cell proliferation, recent studies indicate that Ki67 decays slowly after cell division, and thus can serve as measure of time spent in G0 (*Hogan et al., 2013*; *Miller et al., 2018*; *Sobecki et al., 2017*). In addition, CD8 T cells from intact *Timd4*$^{-/-}$ mice also exhibit slightly elevated levels of Ki67 and have an increased proportion of cells with an activated phenotype. These changes are consistent with increased homeostatic proliferation characteristic of T cells with elevated self-reactivity (*Ge et al., 2004*; *Ge et al., 2001*; *Hogan et al., 2013*). Interestingly, these changes were not observed in CD4 T cells (*Figure 5*). This is consistent with evidence that thymocyte death has a larger impact on the CD8, compared to the CD4, T cell lineage (*Sinclair et al., 2013*) and the relatively modest role of deletion in maintaining tolerance within CD4 lineage T cells, due to the diversion of self-reactive CD4 T cells into the regulatory T cell lineage (*Legoux et al., 2015*; *Malhotra et al., 2016*). Notably, *Timd4*$^{-/-}$ mice do not develop overt autoimmunity, likely due to the presence of other tolerance mechanisms, such as regulatory T cells and peripheral tolerance mechanisms.

Our current data contribute to emerging evidence that the context in which an autoreactive thymocyte encounters peptides, shaped largely by the characteristics of the peptide-presenting cell, has profound impacts on its fate. We recently reported that thymic dendritic cells that provide both high-affinity TCR ligands and a local source of IL-2 can efficiently support the development of regulatory T cells (*Klein et al., 2018*; *Weist et al., 2015*). Thus, the decision of an autoreactive thymocyte to die or differentiate may ultimately depend on whether it engages a peptide-presenting cell that promotes its death or supports its further development.

## Materials and methods

### Mice

All mice were bred and maintained under pathogen-free conditions in an American Association of Laboratory Animal Care-approved facility at the University of California, Berkeley. All procedures were approved by The University of California, Berkeley Animal Use and Care Committee under Animal Use Protocol #AUP-2016-07-9006-1. C57BL/6, C57BL/6-Tg(Ins2-TFRC/OVA)296Wehi/WehiJ (RIPmOVA), C567BL/6-Tg(Csf1r-EGFP-NGFR/FKBP1A/TNFRSF6)2Bck/J (MAFIA), and B6.129S2-Tcra (tm1Mom)/J (*Tcra*$^{-/-}$) mice were from Jackson Labs. OT-I *Rag2*$^{-/-}$ mice were from Taconic Farms. LysMGFP, F5 *Rag1*$^{-/-}$, and *Timd4*$^{-/-}$ mice have been previously described (*Faust et al., 2000*; *Mamalaki et al., 1992*; *Wong et al., 2010*). LysMGFP RIPmOVA and *Timd4*$^{-/-}$ RIPmOVA mice were generated by crossing LysMGFP or *Timd4*$^{-/-}$ mice to RIPmOVA mice. Mice were used from four to eight weeks of age.

### Experimental design

No statistical method of sample size computation was used; 3–6 thymic slices per condition were used in thymic slice experiments, consistent with previous studies. Samples with low viability of the thymic slice or with very low proportions of overlaid thymocytes that had entered the slice were excluded. Experiments where no trend towards negative selection was observed in control conditions were excluded. For experiments where thymic slices from the same genotypic background received multiple treatments, slices were randomly allocated to treatment conditions.

## Thymocyte isolation and labeling

Thymuses were collected from OT-I $Rag2^{-/-}$, F5 $Rag1^{-/-}$, or B6 mice and dissociated through a 70 μm cell strainer to yield a cell suspension. Thymocytes were then labeled with 1 μM Cell Proliferation Dye eFluor450 or 0.5 μM Cell Proliferation Dye eFluor670 (Thermo Fisher Scientific) at $10^7$ cells/ml at 37°C for 15 min in PBS, then washed and resuspended in complete RPMI (containing 10% FBS, penicillin streptomycin, and 2-mercaptoethanol, cRPMI) for overlay onto thymic slices. Thymocytes do not proliferate at the timepoints collected, allowing overlaid thymocytes to be distinguished from slice resident thymocytes by Cell Proliferation Dyes (*Figure 1a*). In imaging experiments, OT-I thymocytes were depleted of mature CD8 single positives using the EasySep Biotin Positive Selection Kit (Stemcell Technologies) with anti-human/mouse β7 integrin antibody (FIB504, Biolegend) according to the manufacturer's instructions. Thymocytes were then labeled with 3 μM SNARF (Thermo Fisher Scientific) at $10^7$ cells/ml at 37°C for 15 min in PBS, then washed and labeled with 5 μM Hoechst 33342 (Thermo Fisher Scientific) at $10^7$ cells/ml at 37°C for 15 min. For all other experiments, the total, unfractionated thymocyte population was used.

## Thymic slices

Preparation of thymic slices has been previously described (*Dzhagalov et al., 2012*; *Ross et al., 2016*). Thymic lobes were cleaned of connective tissue, embedded in 4% agarose with a low melting point (GTG-NuSieve Agarose, Lonza), and sectioned into slices of 200–400 μm using a vibratome (1000 Plus sectioning system, Leica). Slices were overlaid onto 0.4 μm transwell inserts set in six well tissue culture plates with 1 ml cRPMI under the insert. $0.5-2 \times 10^6$ thymocytes in 10 μl cRPMI were overlaid onto each slice and allowed to migrate into the slice for 2 hr, then excess thymocytes were removed by gentle washing with PBS. Thymocytes actively migrate into the slice and localize as expected based on their maturation status (*Ehrlich et al., 2009*; *Halkias et al., 2013*; *Kurd and Robey, 2016*). For peptide-induced negative selection, 10 μl of 1 μM SIINFEKL (AnaSpec) in PBS was overlaid onto each slice for 30 min, then removed by pipetting. To quantify negative selection, we used a fluorescent live/dead stain (Ghost Dye Violet 510 or Fixable Viability Dye eFluor780) to identify live cells (as shown in *Figure 1a*). We then calculated the ratio of total live OT-I thymocytes to total live reference (either wild type thymocytes or thymocytes expressing an irrelevant TCR: F5) recovered from the thymic slice. In general, the ratio of OT-I to reference populations was close to 1.0 in the absence of OVA, however, there was some variability due to differential ability of the two populations to enter or survive in the tissue. We therefore further normalized the ratios of OT-I:reference thymocytes in each experiment so that the average of the corresponding 'no OVA' samples was always 1.0. For depletion of phagocytes, 1 μM AP20187 (Clontech) was added to the media under the transwell and 10 μl of 10 μM AP20187 in PBS was added on top of each slice overnight (16–18 hr). The drug was washed out from the top of the slice with PBS prior to overlaying thymocytes. For Annexin V treatment, thymocytes were resuspended in Annexin V binding buffer (Thermo Fisher Scientific) with purified Annexin V (BioLegend) at 200 μg/ml prior to overlaying on the slice. Following peptide treatment, 10 μl of purified Annexin V at 200 μg/ml in Annexin V binding buffer was overlaid onto each slice.

## Bone marrow-derived dendritic cell cultures

Bone marrow was flushed from the femurs and tibias of mice into sterile PBS, and treated with ammonium chloride–potassium bicarbonate buffer for lysis of red blood cells. Cells were resuspended at $10^6$/ml in cRPMI with 20 ng/ml granulocyte-macrophage colony-stimulating factor (GM-CSF, Peprotech) and plated for culture. Cells were cultured for 7 days, with replacement with fresh media containing GM-CSF on day 6. On day 7, semi-adherent cells were collected and loaded with 1 μM SIINFEKL at $10^7$/ml in cRPMI at 37°C for 30 min. Some BMDCs were incubated without peptide, as indicated. Cells were then washed and $10^5$ BMDCs were overlaid per slice, following washout of excess thymocytes.

## Flow cytometry

Thymic slices, whole thymuses, and spleens were dissociated into FACS buffer (0.5% BSA in PBS) and filtered before staining. Splenocytes and blood samples were treated with ammonium chloride-potassium bicarbonate buffer for 5 or 10 min, respectively, at room temperature prior to staining to

lyse red blood cells. Cells were stained for 10 min on ice in 24G2 supernatant containing the following antibodies: CD4 (GK1.5), CD8α (53–6.7), CD69 (H1.2F3), CD44 (IM7), CD62L (MEL-14). Cells were then washed in PBS and stained in Ghost Dye Violet 510 (Tonbo Biosciences) or Fixable Viability Dye eFluor780 (Thermo Fisher Scientific) for 10 min on ice. For intracellular staining, cells were then fixed and permeabilized using the Transcription Factor Staining Buffer Set (Thermo Fisher Scientific) according to manufacturer's instructions and stained with Ki67 (SolA15) antibody (Thermo Fisher Scientific). For staining of thymic phagocyte populations, whole thymuses were minced and incubated in cRPMI containing 1 mg/ml collagenase Type IA (Sigma) and 400 µg/ml DNase I (Roche) at 37°C for 1 hr. After vigorous pipetting, samples were filtered, then stained in 24G2 supernatant containing the following antibodies: CD11b (M1/70), CD11c (N418), F4/80 (BM8), Tim-4 (RMT4-54), MHC I H-2Kd/H2-Dd (34-1-2S), MHC II I-A/I-E (M5/114.15.2), CD80 (16-10A1), CD86 (GL1), ICAM (YN1/1.7.4). Cells were then washed in PBS and stained in Fixable Viability Dye eFluor780 (Thermo Fisher Scientific) for 10 min on ice. All antibodies were from Thermo Fisher Scientific, Biolegend, or Tonbo Biosciences. Flow cytometry was performed with a LSRII or Fortessa X20 (BD Biosciences) and FlowJo software (TreeStar) was used for data analysis. Gating strategies are shown for thymocytes (*Figure 1a*) and thymic phagocyte populations (*Figure 1—figure supplement 1*).

## Adoptive transfers

Thymuses were isolated from WT or *Timd4*[-/-] mice, dissociated and filtered through a 70 µm cell strainer to yield a cell suspension. $5 \times 10^6$ total WT or *Timd4*[-/-] thymocytes were injected i.v. into *Tcra*[-/-] hosts. Blood was collected at 9–10 weeks post-transfer and analyzed by flow cytometry.

## Two-photon microscopy

Two-photon imaging of thymic slices has been described previously (*Dzhagalov et al., 2013*; *Dzhagalov et al., 2012*; *Melichar et al., 2013*). Briefly, thymic slices were glued to coverslips and fixed to a dish being perfused at a rate of 1 ml/minute with oxygenated, phenol red-free DMEM media warmed to 37°C. Imaging was performed with a Zeiss 7 MP two-photon microscope with a Coherent Chameleon laser tuned to 920 nm. Signals were separated using 495 nm and 560 nm dichroic mirrors. Imaging volumes were scanned every 30 s for 30 min, and images were processed with Imaris 7.3 software (Bitplane).

## Statistics

Statistical analysis was carried out using Prism software (GraphPad). Specific statistical tests used are indicated in figure legends. P values of $< 0.05$ were considered significant.

## Acknowledgements

We would like to thank members of the Robey lab for technical advice and helpful discussion, H Melichar, B Weist, and BJ Fowlkes for critical reading of the manuscript, O Guevarra for technical assistance, P Herzmark for help with two-photon imaging, and Wenjun Ouyang (Genentech) for providing Tim4[-/-] mice. Supported by the National Institutes of Health (R01AI064227 to EAR) and a University of California Cancer Research Coordinating Committee Fellowship to NSK.

## Additional information

### Funding

| Funder | Grant reference number | Author |
| --- | --- | --- |
| National Institutes of Health | R01AI064227 | Ellen A Robey |
| University of California | Cancer Research Coordinating Committee | Nadia S Kurd |

The funders had no role in study design, data collection and interpretation, or the decision to submit the work for publication.

## Author contributions
Nadia S Kurd, Conceptualization, Formal analysis, Funding acquisition, Investigation; Lydia K Lutes, Ashley R Hoover, Formal analysis, Investigation; Jaewon Yoon, Shiao Wei Chan, Investigation; Ivan L Dzhagalov, Conceptualization, Formal analysis, Investigation; Ellen A Robey, Conceptualization, Formal analysis, Supervision, Funding acquisition

## Author ORCIDs
Ellen A Robey  https://orcid.org/0000-0002-3630-5266

## Ethics
Animal experimentation: All mice were bred and maintained under pathogen-free conditions in an American Association of Laboratory Animal Care-approved facility at the University of California, Berkeley. All procedures were approved by the University of California, Berkeley Animal Use and Care Committee under Animal Use Protocol #AUP-2016-07-9006-1.

## Decision letter and Author response
Decision letter https://doi.org/10.7554/eLife.48097.sa1
Author response https://doi.org/10.7554/eLife.48097.sa2

# Additional files

## Supplementary files
• Transparent reporting form

## Data availability
Data that support the findings of this study are reported in figures accompanying the main text and supplementary figures.

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
