## [Decision Letter]

**Acceptance summary:**

By using a thymic tissue slice system, this report demonstrates the impact of phagocytosis on negative selection of T-cells. It also shows that either blocking phosphatidylserine or one of its receptors reduces thymocyte death by affecting phagocytosis. The results facilitate our understanding of the cellular and molecular basis of thymic negative selection, which is important for the establishment of immunological self-tolerance.

**Decision letter after peer review:**

Thank you for submitting your article "A role for phagocytosis in inducing cell death during thymocyte negative selection" for consideration by *eLife*. Your article has been reviewed by three peer reviewers, one of whom is a member of our Board of Reviewing Editors, and the evaluation has been overseen by Satyajit Rath as the Senior Editor. The reviewers have opted to remain anonymous.

The reviewers have discussed the reviews with one another and the Reviewing Editor has drafted this decision to help you prepare a revised submission.

Summary:

During T cell development, autoreactive thymocytes undergo apoptotic cell death induced by strong signals perceived through the T cell receptor (TCR). A number of distinct antigen presenting cell (APC) subsets present self-peptide on MHC and have been shown to facilitate this negative selection process. However, it has been shown that distinct APC subsets may promote different T cell developmental fates independent of TCR-peptide-MHC interactions. Here, Kurd et al. investigate additional cellular interactions that may facilitate thymocyte negative selection. Through the use of a thymic slice system, the authors demonstrate that thymic phagocytes aid in the removal of autoreactive thymocytes following clonal deletion in a phosphatidylserine-Tim-4 dependent manner. Thus, they propose a two-step model by which thymocytes first undergo high affinity self-peptide-MHC interactions, inducing phosphatidylserine on the cell surface. Second, uptake dependent on phosphatidylserine expression leads to subsequent cell death.

Essential revisions:

1) The authors suggest that phagocytosis in the thymus not only serves the purpose of removing dying cells, but in fact is an essential prerequisite for efficient negative selection. This is an extremely interesting and conceptually striking scenario. A major concern is that these conclusions are entirely based on (undoubtedly striking!) observations in the thymic slice model. However, it remains somewhat unclear in how far the thymic slice model reflects steady state clonal deletion. Evidence to support the authors' conclusions in a steady model of in vivo clonal deletion is unfortunately not provided in the manuscript (see comment to Figure 4; this is such an obvious thing to do) and is of crucial importance to meet the standards of *eLife*. The conclusion that phagocytosis mediates negative selection in vivo is perhaps overstated given that the scope of their work is entirely within the slice model system.

2) The reference population seems to be an unspecified mix of irrelevant TCR and polyclonal thymocytes. Individually these are relevant controls, but it is difficult to understand the rational for mixing them together. Evidence that these two populations have similar performance in the assay is required to justify this approach.

3) Figure 1: It would help the reader to better describe the composition of input (and remaining) cells in order to put into perspective that at best 'only' half of the OT-I cells are lost. Are the input cells CD8 SP cells, total cells from OT-I thymi including DP / DN cells or otherwise fractionated cells?

Wouldn't MAFIA + OVA without AP20187 (instead of WT with AP20187) be a necessary control for comparison to MAFIA + OVA with AP20187 in Figures1B and 1C?

4) Figure 1 (and other figures describing CD69 expression): The interpretation of unchanged CD69 expression of cells crucially relies on knowing at which time point this analysis was done. This information seems not to be included. Analyzing this at 16h is probably not a useful indication of what happens to cells that are prone to die before the 16h timepoint. (Are cells that have not died by 16h resistant to peptide induced cell death; see Figure 2? Which again relates to better describing the input population and the composition of cells that remain at 16h: is it preferentially DP cells that are lost?)

5) In addition, in Figure 1B and 1C the WT and WT+AP20187 groups show a trend to difference (every data point is lower in the AP20187 group) even if this does not reach statistical significance. It is also worth noting that many of the OT-I/ref graphs combine experiments to give n numbers in the range of 15-34 while the CD69 MFI graphs have n numbers of 5 and also use a non-parametric test with considerably lower power than the ANOVA applied to the OT-I/ref data. As a result, the statistical tests used on the CD69 MFI graphs are only able to detect much larger effect sizes than those of the OT-I/ref data and should be interpreted with great caution. In particular Figure 1C does not clearly demonstrate that CD69 is not reduced by AP20187 and combining the data from the un shown representative experiments should be considered.

6) Figure 2B: This shows an isolated event. Quantifying the fraction of all recorded death events occurring (a) 'in isolation', (b) in contact and (c) inside a GFP^+^ cell seems necessary to bolster the authors' claim that death and phagocytosis is closely coupled.

7) Figure 3: These are striking observations, but again: without knowing what the input population consists of, it seems surprising that the extent of death (as inferred from a change in the ratio of OT-I to ref cells) is so marginal (< 2-fold) (in particular when OVA peptide is used). In how far was this system tuned towards low doses of OVA peptide, or in other words, does the apparent rescue from apoptosis only occur when low doses of OVA peptide are used? This would be informative also in light of the question whether phagocytosis may be particularly (or only?) crucial in cases of relatively 'low' signal strengths.

8) Figure 4: The authors must have crossed RIPmOVA (*Timd4^-/-^* or *Timd4^+/+^*) to OT-I. This would be extremely important in order to show a role for Tim-4 in 'steady state' negative selection independent of the thymic slice system.

9) Currently the reduction in OT-I cells in Figures 4 and 5 is considered to be due to a loss of phagocytosis but a defect in phagocytosis is not demonstrated in their system. The authors demonstrate that DCs have phagocytic activity, it should be shown that this activity is reduced in Tim-4 KO DCs.

10) The authors propose that direct peptide presentation by a phagocyte results in more efficient negative selection (Figure 5). However, a thymic slice system in which phagocytes lack peptide presenting capability (MHC I), but are still capable of phagocytosis, is an experiment that could more robustly support this claim. The authors demonstrate in Figure 1 that thymocytes still receive strong signals through their TCR (based on CD69 MFI) in the absence of phagocytes, therefore direct peptide presentation by phagocytes may not be necessary for efficient negative selection.

11) In the Discussion, the authors point to several publications that report that cells may initiate early apoptosis programs (active caspase-3 and phosphatidylserine expression), but ultimately avoid cell death. The authors draw a similar conclusion here, but do not show that the thymocytes that would have otherwise undergone phagocytosis in this system are in fact viable and would go on to differentiate. The authors suggest that phagocytes may be the critical last step to thymocyte death, however they should discuss the possibility that phagocytes could just facilitate the efficient removal of dying cells and cell death may ultimately just be delayed in their absence.

12) Figure 1—figure supplement 2B. Control groups showing WT and MAFIA cells without AP20187 are needed.

---

## [Author Response]

Essential revisions:1) The authors suggest that phagocytosis in the thymus not only serves the purpose of removing dying cells, but in fact is an essential prerequisite for efficient negative selection. This is an extremely interesting and conceptually striking scenario. A major concern is that these conclusions are entirely based on (undoubtedly striking!) observations in the thymic slice model. However, it remains somewhat unclear in how far the thymic slice model reflects steady state clonal deletion. Evidence to support the authors' conclusions in a steady model of in vivo clonal deletion is unfortunately not provided in the manuscript (see comment to Figure 4; this is such an obvious thing to do) and is of crucial importance to meet the standards of eLife. The conclusion that phagocytosis mediates negative selection in vivo is perhaps overstated given that the scope of their work is entirely within the slice model system.

To address the in vivo relevance of our observations, we have performed additional analyzes of T cells from *Timd4^-/-^* mice (Figure 5, Figure 5—figure supplement 1, and subsection “The phosphatidylserine receptor Tim-4 promotes negative selection of CD8 T cells in the thymus”, and Discussion, paragraph six). We observe a significant increase in the% of spontaneously activated CD8 T cells in *Timd4^-/-^* mice. This is in line with previous reports that *Timd4^-/-^* mice display elevated anti-DNA antibodies and T cell hyperactivity following immunization (Rodriguez-Manzanet et al., 2010). While these mild autoimmune phenotypes were attributed to defects in peripheral phagocytosis (Rodriguez-Manzanet et al., 2010), our new data show that CD8 T cells that arise in a Tim-4-deficient thymus are intrinsically less quiescent compared to CD8 T cells that develop in wild type mice, even in a Tim-4-sufficient periphery. These new results support our conclusion that Tim-4-mediated phagocytosis promotes efficient negative selection, and in its absence T cells with borderline self-reactivity can escape deletion and survive long-term in the periphery. We have also edited our wording throughout the text to reflect that phagocytosis may not be an absolute requirement for thymocyte death during negative selection, but rather that phagocytosis promotes thymocyte death and increases the efficiency of negative selection (Introduction final paragraph, subsection “The phosphatidylserine receptor Tim-4 promotes negative selection of CD8 T cells in the thymus”, and Discussion paragraph one).

2) The reference population seems to be an unspecified mix of irrelevant TCR and polyclonal thymocytes. Individually these are relevant controls, but it is difficult to understand the rational for mixing them together. Evidence that these two populations have similar performance in the assay is required to justify this approach.

We regret any lack of clarity about our Materials and methods. We did not mix thymocytes with irrelevant TCR (F5 thymocytes) and polyclonal thymocytes (WT thymocytes) together as a reference population. Rather, after verifying that the choice of reference population did not impact the extent of negative selection measured (Author response image 1), we used either WT or F5 thymocytes as a reference population in each experiment depending on availability of mice. We have attempted to clarify this in the revised manuscript (Results section).

**Author response image 1. respfig1:** No significant difference in quantitation of negative selection when F5 versus WT thymocytes are used as an irrelevant reference control. OT-I and either F5 or WT thymocytes were overlaid onto WT thymic slices with or without addition of OVA peptide. 16 hours later, thymic slices were harvested for flow cytometric analysis. Negative selection is displayed as the ratio of live OT-I thymocytes relative to live F5 or WT reference thymocytes, as indicated, normalized to no OVA controls. ns not significant (p<0.05).

3) Figure 1: It would help the reader to better describe the composition of input (and remaining) cells in order to put into perspective that at best 'only' half of the OT-I cells are lost. Are the input cells CD8 SP cells, total cells from OT-I thymi including DP / DN cells or otherwise fractionated cells?

We regret the lack of clarity about our Materials and methods, and have attempted to describe this more clearly in the revised manuscript (Results and Discussion sections). To summarize, we used total, unfractionated, OT-I thymocytes from TCR transgenic H2b mice (positive selecting background) composed mostly of DP and CD8 SP thymocytes for all experiments in which we quantified the extent of OT-I thymocyte deletion. As mentioned in point 4, we did not attempt to separately quantify the loss of DP vs. CD8SP, because antigen encounter by DP thymocytes leads to a rapid downregulation of CD4 and CD8 (a phenomenon termed CD4/CD8 “dulling”)(McGargill and Hogquist, 1999), making it difficult to determine the separation between DP and CD8 SP populations for OT-I thymocytes exposed to OVA. For the imaging experiments (Figure 2B and videos), we depleted the most mature CD8 SP (Author response image 2) prior to adding OT-I thymocytes to the slice. This was done in order to ensure that imaging studies did not include interactions between fully mature CD8 thymocytes and thymic APCs.

**Author response image 2. respfig2:** Depletion strategy used for experiments in Figure 2B and supplemental videos. (**a**) Expression of β7 integrin on the indicated populations of increasing maturity (Hogquist et al., 2015) in wild type and OT-I TCR transgenic thymocytes. (**b**) CD4 vs. CD8 expression on OT-I thymocytes before and after depletion of β7 integrin expressing cells.

Wouldn't MAFIA + OVA without AP20187 (instead of WT with AP20187) be a necessary control for comparison to MAFIA + OVA with AP20187 in Figures 1B and 1C?

Without addition of the dimerization drug AP20187, there is no depletion of phagocytes in MAFIA thymic slices, and accordingly, there is no defect in negative selection as compared to a WT slice (Author response image 3). We did not show this data in the manuscript figure because we felt that it made the figure overly complicated and more difficult to assess the most relevant comparisons, but we have compiled the data from the untreated MAFIA and untreated WT slices in the new Figure 1B and pointed out the fact that there is no difference in negative selection between these two conditions in the revised manuscript (subsection “Depletion of thymic phagocytes inhibits negative selection” paragraph two). We would be happy to separate this data in the manuscript if the reviewers feel it is necessary.

**Author response image 3. respfig3:** No significant difference in negative selection on untreated WT versus MAFIA thymic slices. WT or MAFIA thymic slices were cultured with or without AP20187 for 16 hours. OT-I and reference thymocytes (either wild type or F5 TCR transgenic) were then overlaid onto the thymic slices with or without OVA peptide, and dissociated for analysis by flow cytometry after 16 hours of culture. Thymocyte death is displayed as the ratio of live OT-I thymocytes relative to live reference thymocytes present within the slice, normalized to the average ratio on no antigen control slices. ns not significant (p>0.05), **p<0.01, (Unpaired two-tailed t test). Data are pooled from 3 independent experiments with mean and SEM of n=16 total slices per condition (except for MAFIA+OVAp condition, n=6), where each dot represents an individual slice.

4) Figure 1 (and other figures describing CD69 expression): The interpretation of unchanged CD69 expression of cells crucially relies on knowing at which time point this analysis was done. This information seems not to be included. Analyzing this at 16h is probably not a useful indication of what happens to cells that are prone to die before the 16h timepoint. (Are cells that have not died by 16h resistant to peptide induced cell death; see Figure 2? Which again relates to better describing the input population and the composition of cells that remain at 16h: is it preferentially DP cells that are lost?)

The levels of CD69 shown in the figures are from the same samples in which death is measured, and were cultured for 16 hours. By measuring CD69 levels at 16 hours, our goal was to focus on thymocytes that had survived negative selection under conditions of impaired phagocytosis, in order to rule out that these cells escaped death due to insufficient TCR signals. We have reworded the relevant sections of the figure legends (for Figures 1C, 3B, 3D, 4B, 4D, and 6C) and Results section to clarify this point. We are currently preparing a separate manuscript describing the fate of thymocytes that survive negative selection after 16 hours, which provides evidence that these thymocytes undergo agonist selection into the CD8αα intraepithelial lymphocyte lineage, and describes factors which may protect some thymocytes from death and instead promote agonist selection into this lineage. As mentioned in point 3, we did not attempt to separately quantify the loss of DP vs. CD8SP, because antigen encounter by DP thymocytes leads to a rapid downregulation of CD4 and CD8 (a phenomenon termed CD4/CD8 “dulling”)(McGargill and Hogquist, 1999), making it difficult to determine the separation between DP and CD8 SP populations on slices where thymocytes are exposed to OVA.

5) In addition, in Figure 1B and 1C the WT and WT+AP20187 groups show a trend to difference (every data point is lower in the AP20187 group) even if this does not reach statistical significance. It is also worth noting that many of the OT-I/ref graphs combine experiments to give n numbers in the range of 15-34 while the CD69 MFI graphs have n numbers of 5 and also use a non-parametric test with considerably lower power than the ANOVA applied to the OT-I/ref data. As a result, the statistical tests used on the CD69 MFI graphs are only able to detect much larger effect sizes than those of the OT-I/ref data and should be interpreted with great caution. In particular Figure 1C does not clearly demonstrate that CD69 is not reduced by AP20187 and combining the data from the un shown representative experiments should be considered.

While there is a slight trend towards reduced death in the WT+AP20187+OVAp group in Figure 1B, there is a clear significant difference between this group and the MAFIA+AP20187+OVAp, suggesting that treatment with drug has the biggest impact on MAFIA slices, in which phagocytes are susceptible to deletion, and any potential off-target impacts of the drug on negative selection are relatively minimal. Moreover, our findings that negative selection is also impaired upon two additional conditions in which phagocytosis is impaired (blocking phosphatidylserine and in a Tim-4 deficient thymic environment), strongly suggest that the impairment in negative selection that we observe on MAFIA+AP20187+OVAp slices is driven mainly by a lack of phagocytosis as a result of phagocyte depletion, and not an off-target effect of the drug.

To provide a more robust measurement of CD69 upregulation, we have now combined the CD69 data from all independent experiments in Figure 1C, and in all additional figures demonstrating CD69 upregulation, and applied the same ANOVA statistical test that we applied to the OT-I/ref graphs (Figures 1C, 3B, 3D, 4B, 4D, and 6C, and corresponding figure legends).

6) Figure 2B: This shows an isolated event. Quantifying the fraction of all recorded death events occurring (a) 'in isolation', (b) in contact and (c) inside a GFP^+^ cell seems necessary to bolster the authors' claim that death and phagocytosis is closely coupled.

We have previously published an imaging study of negative selection to a ubiquitous self-antigen (peptide-induced)(Dzhagalov et al., 2013) in which we provided extensive quantification of cell death and phagocytosis. To summarize the relevant findings from the earlier study: we observed a total of 31 cell death events of which all 31 occurred while the dying thymocyte was in close contact with, or engulfed by, a phagocyte. In comparison, of the thymocytes from these same runs that did not die, 54% were in contact with, and 17% were engulfed by, a phagocyte. We also reported that for 14 examples of cell death (34%) we were also able to observe the initial contact with the phagocyte, and that the time between the initial contact with a phagocyte and death varied considerably from 2-56 minutes. We note that many of the experiments in the current manuscript use a peptide induced model of negative selection, so our earlier published imaging studies should provide a valid comparison to the functional studies presented here. We included Figure 2B and the videos only to make the point that simultaneous phagocytosis and death also occurs with AIRE-dependent negative selection.

7) Figure 3: These are striking observations, but again: without knowing what the input population consists of, it seems surprising that the extent of death (as inferred from a change in the ratio of OT-I to ref cells) is so marginal (< 2-fold) (in particular when OVA peptide is used). In how far was this system tuned towards low doses of OVA peptide, or in other words, does the apparent rescue from apoptosis only occur when low doses of OVA peptide are used? This would be informative also in light of the question whether phagocytosis may be particularly (or only?) crucial in cases of relatively 'low' signal strengths.

As mentioned in points 3 and 4, our input population consisted of total OT-1 thymocytes consisting of both DP and CD8SP thymocytes, and we were unable to separately quantify the loss of DP vs. CD8SP, due to CD4/CD8 “dulling” on OVA-containing thymic slices. With regard to the extent of negative selection, this was in line with a previously published study in which thymocytes were depleted of mature CD8 single positive cells prior to overlay on the slice (see Figure 5A from Dzhagalov et al., 2013). Thus it is unlikely that the use of an unfractionated thymocyte population greatly impacted the extent of negative selection in our system. While a relatively high concentration of OVA peptide was used in Figure 3 and 4, we agree with the reviewers point that phagocytosis may play a more important role in inducing thymocyte death under limiting antigen conditions. Indeed, we proposed this explanation for the observation that negative selection to a tissue-restricted antigen, which is present at low abundance, was more greatly impacted by Tim-4 deficiency than negative selection to exogenously added OVA peptide (Figure 4A and C, Discussion paragraph six).

8) Figure 4: The authors must have crossed RIPmOVA (Timd4^-/-^ or Timd4^+/+^) to OT-I. This would be extremely important in order to show a role for Tim-4 in 'steady state' negative selection independent of the thymic slice system.

In the revised manuscript, we provide new in vivo evidence for an impact of Tim-4 deficiency on CD8 T cell thymic development (Figure 5, Figure 5—figure supplement 1, and subsection “The phosphatidylserine receptor Tim-4 promotes negative selection of CD8 T cells in the thymus”, and Discussion paragraph six). While the mild autoimmune phenotype of these mice had been previously attributed to defects in peripheral phagocytosis (Rodriguez-Manzanet et al., 2010), our new data show that CD8 T cells that arise in a Tim-4 deficient thymus are intrinsically less quiescent compared to CD8 T cells that develop in wild type mice, even in a Tim-4-sufficient peripheral environment. These new results support our conclusion that Tim-4-mediated phagocytosis is required for efficient negative selection, and in its absence T cells with borderline self-reactivity can escape deletion. Given that loss of Tim-4 had a modest impact on negative selection compared to Annexin V blocking or phagocyte depletion (likely due to the fact that other PS receptors remain intact in *Timd4^-/-^* animals), and given that there are also multiple tolerance mechanisms still in place (regulatory T cells and other peripheral tolerance mechanisms), we would not expect to observe overt autoimmunity in *Timd4^-/-^* mice.

With regard to the specific experiment suggested by the reviewer, we did not attempt to generate OT-IxRIPmOVAxTim-4^-/-^ mice, since OT-IxRIPmOVA mice already have a high mortality rate shortly after birth (Koehli et al., 2014). As an alternative strategy, we attempted to examine the impact of Tim-4 deficiency on tolerance in the RIPmOVA model by generating neonatal partial hematopoietic chimeras in which OT-I bone marrow was injected into RIPmOVA or Tim-4^-/-^ RIPmOVA neonates. Unfortunately, this experiment was not interpretable due to residual contaminating mature OT-I T cells from the bone marrow which were able to home to the thymus and expand in response to OVA expression there. Since these cells were mature at the time of transfer, they were not subject to negative selection in the thymus, making it impossible to discern whether any differences in negative selection existed between the RIPmOVA and Tim-4^-/-^ RIPmOVA hosts.

9) Currently the reduction in OT-I cells in Figures 4 and 5 is considered to be due to a loss of phagocytosis but a defect in phagocytosis is not demonstrated in their system. The authors demonstrate that DCs have phagocytic activity, it should be shown that this activity is reduced in Tim-4 KO DCs.

Several studies have previously demonstrated that Tim-4 plays an important role in phagocytosis within several phagocyte populations, including thymic phagocytes (Miyanishi et al., 2007; Rodriguez-Manzanet et al., 2010; Wong et al., 2010).

10) The authors propose that direct peptide presentation by a phagocyte results in more efficient negative selection (Figure 5). However, a thymic slice system in which phagocytes lack peptide presenting capability (MHC I), but are still capable of phagocytosis, is an experiment that could more robustly support this claim. The authors demonstrate in Figure 1 that thymocytes still receive strong signals through their TCR (based on CD69 MFI) in the absence of phagocytes, therefore direct peptide presentation by phagocytes may not be necessary for efficient negative selection.

Several previous studies have demonstrated reduced efficiency of negative selection under conditions in which hematopoietic-derived cells (including phagocytes) cannot present antigen, but would be capable of phagocytosis, in line with our findings that antigen presentation by phagocytic cells promotes negative selection (McCaughtry et al., 2008; Melichar et al., 2013; Proietto and Wu, 2008; Wirasinha et al., 2019; Yap et al., 2018). Moreover, we believe that the experiment presented in Figure 5 addresses the same questions as the experiment proposed by the reviewer. In this setting, endogenous phagocytes do not present the antigenic peptide (because the peptide is exclusively loaded onto the APC population that is added to the thymic slice), but retain normal phagocytic activity. Under these conditions, the phagocytic ability of the peptide presenting population significantly impacts the efficiency of negative selection. Moreover, we did not intend to argue that peptide presentation by phagocytes is a requirement for negative selection. Rather, we interpret our data to indicate that the efficiency of negative selection is improved when the same cell can both present agonist peptide and phagocytose the self-reactive thymocytes, as outlined in the Discussion and Figure 6—figure supplement 2.

11) In the Discussion, the authors point to several publications that report that cells may initiate early apoptosis programs (active caspase-3 and phosphatidylserine expression), but ultimately avoid cell death. The authors draw a similar conclusion here, but do not show that the thymocytes that would have otherwise undergone phagocytosis in this system are in fact viable and would go on to differentiate. The authors suggest that phagocytes may be the critical last step to thymocyte death, however they should discuss the possibility that phagocytes could just facilitate the efficient removal of dying cells and cell death may ultimately just be delayed in their absence.

We have added additional text to the Discussion acknowledging that thymocyte death may be delayed, rather than completely blocked, in the absence of phagocytosis (paragraph two). We also note that the revised manuscript contains new in vivo data (Figure 5, Figure 5—figure supplement 1, and subsection “The phosphatidylserine receptor Tim-4 promotes negative selection of CD8 T cells in the thymus”, and Discussion, paragraph five) suggesting enhanced survival of autoreactive thymocytes due to defective thymic phagocytosis. Specifically, we show that CD8 T cells that arise in a Tim-4 deficient thymus have reduced quiescence weeks after residing in a Tim-4 sufficient peripheral environment. These data imply that impaired phagocytosis in the thymus has a long-lasting impact on T cells, likely due to the failure of some T cells with borderline self-reactivity to undergo negative selection.

12) Figure 1—figure supplement 2B. Control groups showing WT and MAFIA cells without AP20187 are needed.

In Figure 1—figure supplement 2, the values shown represent the proportion of viable macrophages or dendritic cells in the slice after 16-18 hours of AP20187 treatment normalized to the proportion of viable macrophages or dendritic cells present within untreated slices, therefore the mean for the untreated groups is set to 1. We have now added these groups for better visualization of the changes that occur in response to AP20187 treatment.